# Preparation Process and Phase Transformation of Al-5Ti-0.25C Master Alloy Adopting Ti Machining Chips

**DOI:** 10.3390/ma14195783

**Published:** 2021-10-03

**Authors:** Sanbo Li, Chunfang Zhao, Fei Wang, Maoliang Hu, Zesheng Ji, Sumio Sugiyama

**Affiliations:** 1School of Mechanical and Electrical Engineering, Lishui Vovational and Technical College, Lishui 323000, China; zhaochunfang0710@126.com; 2School of Materials Science and Chemical Engineering, Harbin University of Science and Technology, Harbin 150001, China; wfambitious@126.com (F.W.); humaoliang@hrbust.edu.cn (M.H.); jizesheng@hrbust.edu.cn (Z.J.); 3Institute of Industrial Science, The University of Tokyo, Tokyo 153-8505, Japan; sugi@iis.u-tokyo.ac.jp

**Keywords:** master alloy, preparation process, grain refinement, wettability

## Abstract

The refining performance of Al-Ti-C master alloys is substantially compromised by the inferior wettability between graphite and molten aluminum. In this paper, the Al-5Ti-0.25C master alloy was successfully prepared by reacting Ti machining chips, graphite, and molten aluminum. In order to determine a simple method of improving the wettability, the optimal preparation process and phase transformation of the Al-5Ti-0.25C master alloy were investigated using an optical microscope, X-ray diffractometer, and scanning electron microscope equipped with an energy dispersive spectrometer. The results show that the feeding method using a prefabricated block made from Ti chips, Al chips, and graphite effectively improves the wettability between graphite and molten aluminum and increases the recovery rate of graphite. When the reaction temperature is low (1223 K), the agglomeration of TiAl_3_ is caused. When the reaction temperature is high (1373 K), the morphology of TiAl_3_ changes from block-like to needle-like and increases its size. Further, a short reaction time (30 min) results in the incomplete dissolution of the Ti chips, while a long reaction time (90 min) causes the TiAl_3_ to transform into needle-like morphologies. The microstructural observation of undissolved Ti chips shows that TiAl_3_ and TiC are formed around it, which proves the transformation of Ti chips to TiAl_3_ and TiC. In addition, the enrichment of TiC and Al_4_C_3_ was observed in the vicinity of TiAl_3_, and a reaction model for the formation of TiC from the reaction of Al_4_C_3_ and TiAl_3_ was presented.

## 1. Introduction

Aluminum alloys are widely used in aerospace, automobiles, and machinery due to their low density, light structure, and excellent mechanical properties. Material development requires higher mechanical properties for aluminum alloys, and grain refinement plays an important role in improving their properties. The addition of master alloys is the most simple and effective method to reduce grain size [1,2,3,4].

Numerous research studies on master alloys for aluminum alloys have made a great amount of progress. One of the most effective elements for the grain refinement of aluminum alloys is the Ti element, which forms Ti-containing intermetallic compounds as effective heterogeneous substrates and has a strong growth restriction factor caused by constitutional undercooling [5,6]. Among the compounds studied, the Al-Ti-B master alloy has been extensively used in industrial production [7]. However, the shortcomings of the Al-Ti-B master alloy have gradually been exposed throughout its extensive applications. For example, TiB_2_—when used as the heterogeneous substrate—has an intense aggregation tendency, which seriously weakens the refining performance. Moreover, the “poisoning effect ” of the Al-Ti-B master alloy occurs in hypereutectic Al-Si and Zr-containing alloys [8,9]; that is, Si and Zr noticeably decrease the effectiveness of grain refinement with the Al-Ti-B refiner.

Since Cibula [10] put forward the carbide boride theory, Al-Ti-C master alloys or multi-component master alloys containing C, are of significant interest to researchers, and tremendous efforts have been devoted to investigating Al-Ti-C master alloys. Fortunately for Al-Ti-C master alloys, a series of master alloys [11,12] have been developed after the Al-Ti-C master alloy was successfully prepared by Banerji and Reif [13,14]. The results show that Al-Ti-C master alloys not only have distinguished refining performance for aluminum alloys but also have the advantage of resisting the “poisoning“ phenomenon for alloys containing Zr and Si [15,16]. Lsiterature [17,18] proposes that the similarity of the crystal structure and nearest-neighbor atomic distance in the relevant contacting planes, such as Ti(Al, Zn)_3_, have an important influence on the nucleation of α-Al. As a brand new advantage, TiC is less prone to agglomeration than TiB_2_, and the lattice mismatch between TiC and α-Al is smaller compared to TiB_2_ [14,19,20,21]. However, poor wettability between graphite and liquid aluminum makes the preparation of A1-Ti-C master alloys difficult and prevents its application in industry. The current hotspot for Al-Ti-C is to solve the problem of wettability and develop an economical preparation process. The machining process produces a large number of waste Ti chips. These chips are difficult to recycle.

It is very important to explore a simple method of recycling Ti machining chips and improve the wettability between graphite and molten aluminum. In this paper, the preparation process of the Al-5Ti-0.25C master alloy is carefully discussed using Ti chips as raw materials. The refining performance of the Al-5Ti-0.25C master alloy is compared with that of the commercial Al-5Ti-1B master alloy.

## 2. Materials and Methods

Al chips (purity ≥ 99.78 wt.%), Ti chips (Ti 90.2 wt.%, Al 6.1 wt.%, and V 3.7 wt.%), and graphite powder (purity ≥ 99.9%, about 30 μm in size) were uniformly mixed and pressed into a prefabricated block (Φ38 mm × 10 mm) at room temperature. The pure aluminum was melted at 1033 K using an RX2-37-13 box-type resistance furnace (Harbin Songjiang Electric Furnace Factory Co., Ltd., Harbin, China) with excellent temperature control accuracy, and it was further heated to reaction temperature for the addition of the prefabricated block. The temperature indicator was submerged in the liquid. When the temperature indicator of the RX2-37-13 box-type resistance furnace displayed the furnace temperature as the preset reaction temperature, the prefabricated block was added into the aluminum melt within a few seconds. The reaction temperatures were 1223 K, 1273 K, and 1373 K. After that, the aluminum melt with the added prefabricated block was placed in the furnace for reaction and kept there for a period of time (reaction time). The reaction times were 30 min, 60 min, and 90 min. The melt was poured into a steel mold (Φ40 mm × 40 mm) to obtain Al-5Ti-0.25C master alloys. The feeding methods were divided into two methods: FW-1, pre-mixed Ti chips, Al chips, and graphite powder externally, added in the form of a prefabricated block (Φ38 mm × 10 mm), which was made by compressing pre-mixed Ti chips, Al chips, and graphite powder using a hydraulic press at room temperature: FW-2, a prefabricated block, made using Al chips and Ti chips, was added first, and then the prefabricated block made using Al chips and graphite powder was added. The reaction temperatures were 1223 K, 1273 K, and 1373 K, respectively. The reaction times were 30 min, 60 min, and 90 min, respectively. The schematic illustration of the preparation process of the Al-5Ti-0.25C master alloy is shown in Figure 1.

The refinement tests were carried out in pure aluminum for the comparison of the refining performance of Al-5Ti-0.25C with that of Al-5Ti-1B. The amount of the master alloy added was 0.2 wt.%. The refining temperature was 1023 K. The refining time was 2 min. The samples of the refined aluminum alloy were polished and etched by a reagent (60 vol.% HCl + 30 vol.% HNO_3_ + 5 vol.% HF + 5 vol.% H_2_O) within 15–20 s.

The metallographic observation was carried out using an Olympus optical microscope (OM) (OLYMPUS-GX71-6230A, Olympus (China) Co., Ltd., Suzhou, China), and the average grain size was evaluated using the linear intercept method. The phases were analyzed using X-ray diffraction (XRD) with Cu-Kα radiation (scanning speed of 10°/min, step size of 0.02°, scanning angle from 20° to 90°). The Fei-quanta 200 scanning electron microscope (SEM), (Fei-quanta 200, FEI Company, Hillsboro, OR, USA) equipped with an energy dispersive spectrometer (EDS) (SystemSix, Thermo Fisher Company, Shanghai, China), was employed to observe the microstructures. The average grain size of all refined samples was measured using the linear intercept method.

## 3. Results and Discussion

### 3.1. Phase Analysis

The XRD pattern and EDS elemental mappings of the FW-1 master alloy at 90 min are shown in Figure 2. Figure 2a shows that the Al-5Ti-0.25C master alloy is mainly composed of TiAl_3_ and TiC, with the average sizes of about 25.2 μm and 2.5 μm, except for the Al matrix; this has also been confirmed by many other researchers who have investigated the master alloy of the Al-Ti-C system [19,20]. The EDS elemental mappings in Figure 2b–e reveal that the needle-like TiAl_3_ and the granular TiC show a huge difference in morphology.

### 3.2. Preparation Process

Figure 3 shows the undissolved graphite of the FW-1 and FW-2 master alloys at 60 min and 1273 K. The presence of more graphite and undissolved graphite in FW-2 compared to FW-1 show that the wettability of graphite and molten aluminum can be improved by adding the prefabricated block. As is well known, the microstructures of the master alloy have a great influence on their refining performance. The existence of undissolved graphite seriously reduces the refining performance. In order to evaluate the feeding methods, the recovery rate of Ti and C were calculated using the method described in detail in the literature [21]. The results demonstrated that the recovery rates of Ti and C in FW-1 are 96% and 59%, respectively, while those in FW-2 are only 94% and 37%, respectively. In terms of feeding method, the FW-1 master alloy, adopting the feeding method of the prefabricated block, has more advantages.

Figure 4 shows the microstructures of FW-1 with respect to different reaction temperatures when the reaction time is 60 min. As evidence by Figure 4a, the serious agglomeration phenomenon of TiAl_3_ is presented in the microstructure at 1223 K. The low reaction temperature leads to weak thermal convection, poor fluidity, and high viscosity of the melt, resulting in TiAl_3_ being difficult to disperse at 1223 K. The distribution of TiAl_3_ is uniform without agglomeration when the reaction temperature is higher than 1273 K, as shown in Figure 4b. However, the morphology of TiAl_3_ changes from block-like to needle-like at 1373 K in large quantities (shown in Figure 4c), thanks to the obvious preferred growth orientation at 1373 K [22]. The block-like TiAl_3_ is an equiaxed crystal, which provides more nucleation base planes and has a faster dissolution rate than other morphologies [20,22,23,24]. It is obvious that 1273 K is the most suitable reaction temperature in this investigation, although the average size of TiAl_3_ at 1273 K is slightly larger than at 1223 K.

Figure 5 shows the microstructures of FW-1 with respect to (a) 30 min, (b) 60 min, and (c) 90 min reaction times at a temperature of 1273 K. As shown in Figure 5a, the reaction time of 30 min cannot drive the complete dissolution of Ti chips and graphite, resulting in partially undissolved graphite and Ti chips remaining in the microstructure of FW-1. Figure 5b reveals that FW-1 possesses an excellent microstructure without undissolved graphite and Ti chips when the reaction time is 60 min. However, when the reaction time is 90 min, the morphology of TiAl_3_ changes from block-like to needle-like (Figure 5c). As illustrated by Figure 5d–f, the average size of the TiAl_3_ phase increases with the increase in reaction time and ranges from 15.26 μm at 30 min to 17.71 μm at 60 min and then to 30.47 μm at 90 min. The secondary electron image of the undissolved Ti chips is shown in Figure 6, which proves TiAl_3_ and TiC are abundant around the undissolved Ti chips. Y Dong [25] also observed the formation of TiAl_3_ at the interface between Ti powder and molten aluminum. In addition, TiC clusters around the Ti chip are also observed in the microstructure.

### 3.3. Phase Transformation

The refining performance of Al-5Ti-0.25C is heavily dependent on TiAl_3_ and TiC. Therefore, it is very important to discuss the stability and transformation of TiAl_3_ and TiC. Fine and Conley [26] pointed out that three-phase equilibrium coexists with TiC, TiAl_3,_ and the liquid phase in the Al-Ti-C ternary system. The formation of TiAl_3_ and TiC may occur through the following reaction formula during the preparation of the Al-5Ti-0.25C master alloy:3Al + Ti = TiAl_3_
(1)
Ti + C = TiC (2)

After adding the prefabricated block, the block is decomposed and slowly melted. Ti chips directly react with molten aluminum to generate TiAl_3_, as shown in Equation (1). The literature [26] demonstrates that the wettability between Al and Ti is greater than that between Al and C; that is to say, the existence of Equation (1) is reasonable. As described in Figure 5, a large amount of TiAl_3_ exists around the undissolved Ti chips.

After Al chips, Ti chips, and graphite powder are pressed as a whole, graphite is deposited on the surface of Al chips and Ti chips under the action of external force; therefore, graphite does not easily float to the surface of the molten aluminum, which effectively reduces the burning loss caused by graphite floating. It has been found that Ti and graphite can react to form TiC at 1273 K [27], as shown in Equation (2). The formation of TiAl_3_ releases a lot of heat and forms the local high temperature, which improves the wettability of graphite and molten aluminum and promotes the formation of TiC [28].

Another point is the enrichment of TiC near TiAl_3_ and the existence of Al_4_C_3_, as illustrated by Figure 7. The instability of TiAl_3_ and Al_4_C_3_ compared to TiC has been proved by numerous studies; therefore, TiC can be formed by reacting Al_4_C_3_ with TiAl_3_ [25,28,29], as proven by Equation (3). The reaction between TiAl_3_ and Al_4_C_3_ has also been confirmed by other researchers [30,31,32].
3TiAl_3_ + Al_4_C_3_= 13Al + 3TiC(3)

Based on the above discussion on the formation of TiC, a schematic illustration of the formation mechanism of TiC resulting from the reaction of Al_4_C_3_ and TiAl_3_ is presented in Figure 8. The dissolution of TiAl_3_ and Al_4_C_3_ in molten aluminum releases Ti atoms and C atoms, respectively. Due to the solubility of C in molten aluminum being extraordinarily limited—only about 3 × 10^−4^ wt.% at 1273 K [28]—the mass consumption of Ti atoms is released by the dissolution of the Ti chips. Consequently, the Ti atoms and C atoms, released from the dissolution of TiAl_3_ and Al_4_C_3_, combine to form TiC.

### 3.4. Refining Performance

The coarse columnar crystal and the equiaxed crystal of pure aluminum, both with an average grain size of about 3000 μm, are observed before being added to the master alloy. Figure 9 reveals the macrostructure and microstructure of pure aluminum refined by the Al-5Ti-0.25C and Al-5Ti-1B master alloys. The results of refinement tests show that the average grain size of pure aluminum is significantly refined, and the grains become fine equiaxed ones after being added to the Al-5Ti-0.25C master alloy. The average grain size of pure aluminum is refined to about 190 μm, which is lower than the 250 μm size of Al-5Ti-1B.

## 4. Conclusions

The optimum preparation process of the Al-5Ti-0.25C master alloy is the feeding method of adopting the prefabricated block made by Ti chips, Al chips, and graphite. The reaction temperature and reaction time are 1273 K and 60 min, respectively.

TiAl_3_ and TiC, as effective heterogeneous nucleation sites, are formed around Ti chips. Since TiAl_3_ and Al_4_C_3_ are more thermodynamically unstable than TiC, TiC can be formed through the reaction of TiAl_3_ and Al_4_C_3_.

The comparison of the refining performances between the Al-5Ti-0.25C and Al-5Ti-1B master alloys shows that the Al-5Ti-0.25C master alloy, fabricated through the machining of Ti chips, is superior. The average grain size of pure aluminum refined by the Al-5Ti-0.25C master alloy is significantly refined to about 190 μm, which is smaller than the 250 μm size of Al-5Ti-1B.

## Figures and Tables

**Figure 1 materials-14-05783-f001:**
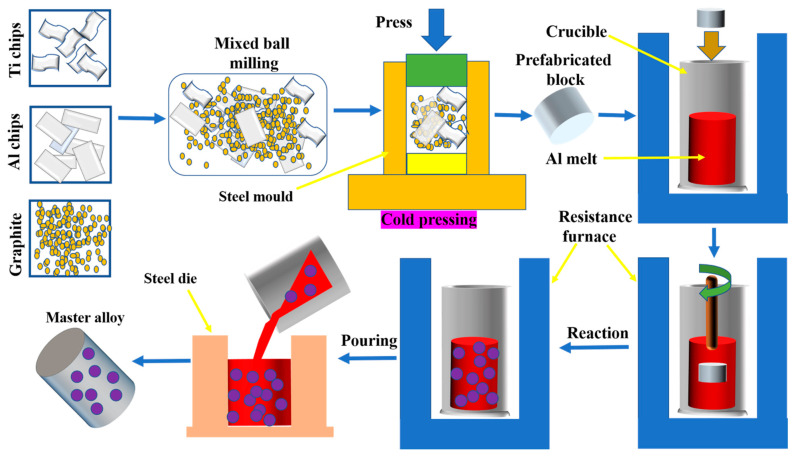
Schematic illustration of the preparation process of the Al-5Ti-0.25C master alloy.

**Figure 2 materials-14-05783-f002:**
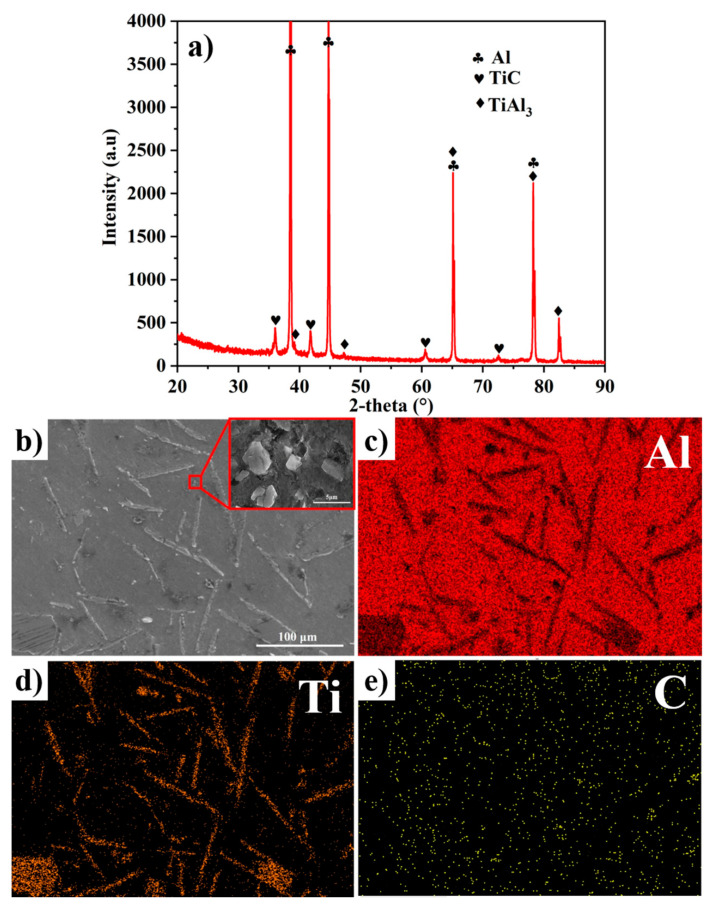
(**a**) The XRD pattern and (**b**–**e**) EDS elemental mapping of FW-1 at 90 min.

**Figure 3 materials-14-05783-f003:**
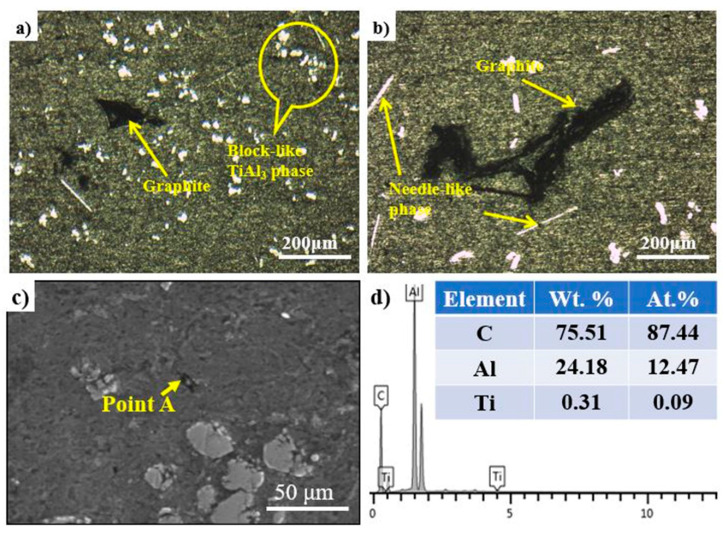
The undissolved graphite of the FW-1 and FW-2 master alloys at 60 min and 1273 K: (**a**) FW-1; (**b**) FW-2; (**c**) a secondary electron image of the undissolved graphite in FW-1; and (**d**) the EDS analysis of point A marked in (**c**).

**Figure 4 materials-14-05783-f004:**
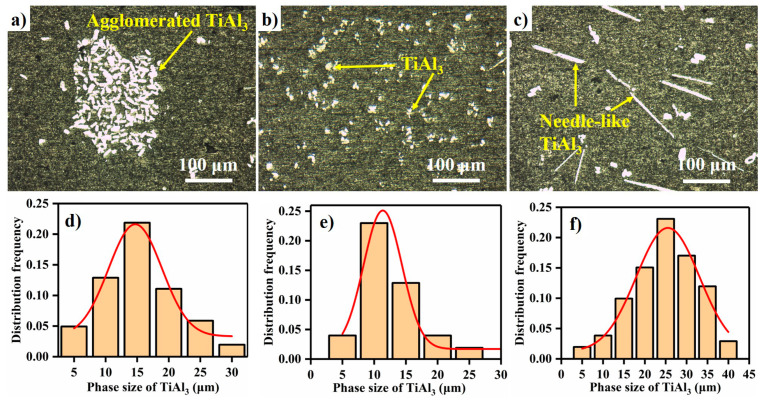
Microstructures of FW-1 at temperatures of: (**a**) 1223 K, (**b**) 1273 K, and (**c**) 1373 K, under the reaction time of 60 min; and (**d**–**f**) the size-distribution frequency statistics of TiAl_3_, corresponding to temperatures of 1223 K, 1273 K, and 1373 K, respectively.

**Figure 5 materials-14-05783-f005:**
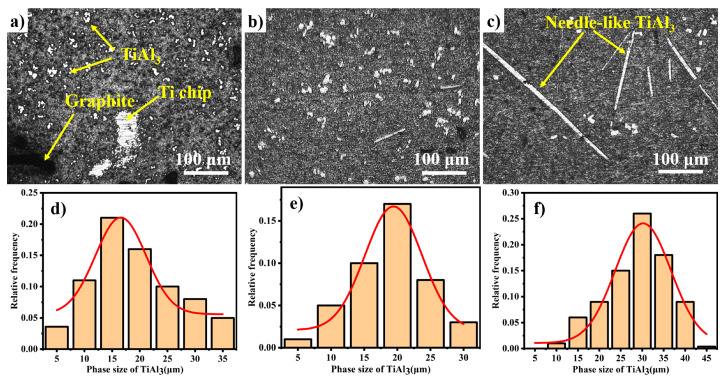
Microstructures of FW-1 with respect to (**a**) 30 min, (**b**) 60 min, and (**c**) 90 min reaction times at a temperature of 1273 K; and (**d**–**f**) the size-distribution frequency statistics of TiAl_3_, corresponding to 30 min, 60 min, and 90 min reaction times, respectively.

**Figure 6 materials-14-05783-f006:**
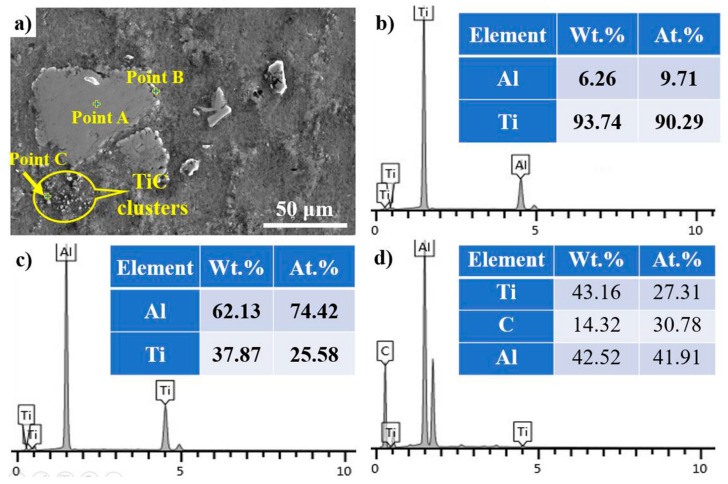
(**a**) The secondary electron image of undissolved Ti chips; and the EDS spectrums of (**b**) point A, (**c**) point B, and (**d**) point C.

**Figure 7 materials-14-05783-f007:**
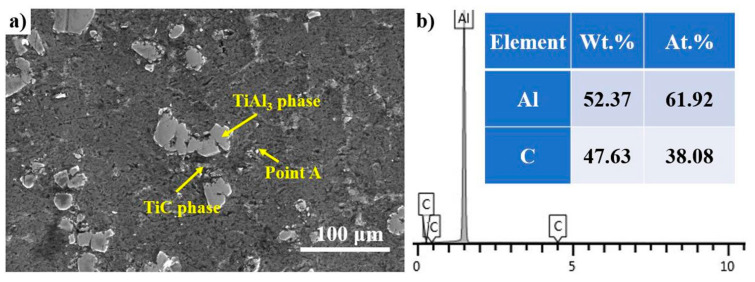
(**a**) Secondary electron image of TiAl_3_ and Al_4_C_3_; and (**b**) the EDS spectrums of point A.

**Figure 8 materials-14-05783-f008:**
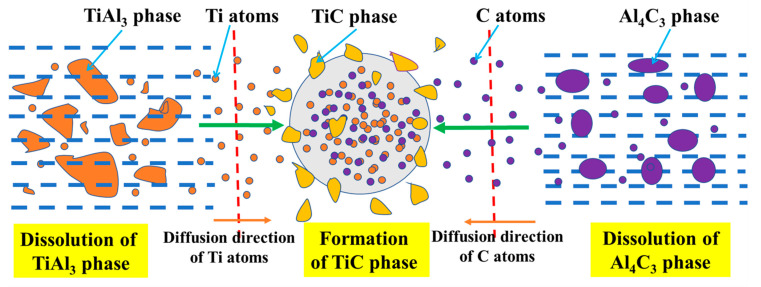
Schematic illustration of the formation of TiC from the reaction of Al_4_C_3_ and TiAl_3_.

**Figure 9 materials-14-05783-f009:**
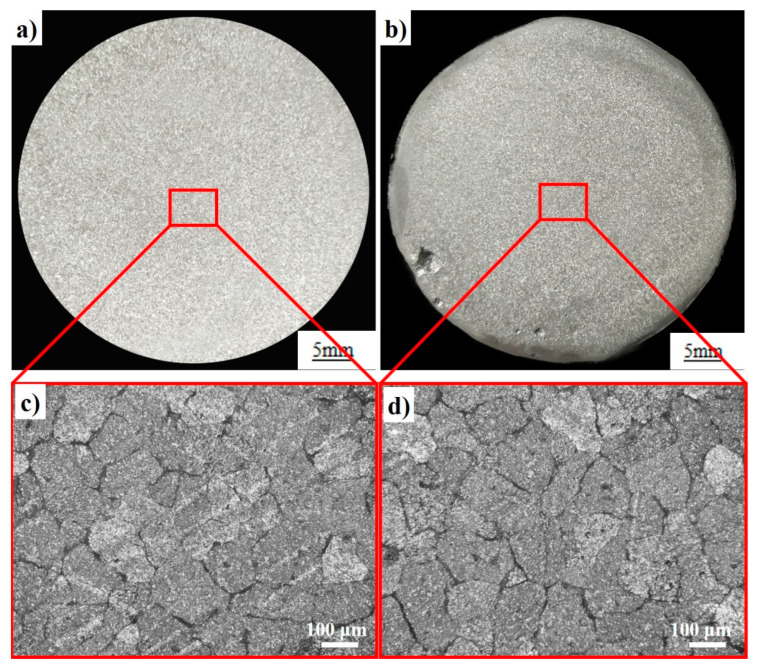
Macrostructures of pure aluminum refined by the (**a**) Al-5Ti-0.25C and (**b**) Al-5Ti-1B master alloys; (**c**) the magnified microstructure of (**a**); and (**d**) the magnified microstructure of (**b**).

## Data Availability

Data available in a publicly accessible repository.

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
