# Peer review of "Preparation Process and Phase Transformation of Al-5Ti-0.25C Master Alloy Adopting Ti Machining Chips"

_materials, 2021, doi:10.3390/ma14195783_

Round 1
Reviewer 1 Report
The work “Preparation process and phase transformation of Al-5Ti-0.25C master alloy adopting Ti machining chips” can be reconsidered after major revision. The comments are drawn as follows:
The number of lines is missed in the pdf file, please, mention this when resubmitting the manuscript.
- “The results show that Al-Ti-C master alloys not only have distinguished refining performance for aluminum alloys but also have the advantages of resistance to Zr- and Si-containing alloys, because TiAl3 and TiC have an important influence on grain refinement [15, 16]” what do authors mean under resistance to Zr- and Si-containing alloys?
- Authors write: “The reaction temperature was 1223 K, 1273 K, and 1373 K, respectively.” (1) respectively to what? (2) How was the reaction temperature measured? If it is the exposure temperature before the casting it cannot be described by the term “reaction”.
- “The results dominate that the recovery rate of Ti and C in FW-1 is 96% and 59%, respectively, while that in FW-2 is only of 94% and 37%, respectively.” How were these data calculated?
- Figures 3 (c) and (d) are not described in the caption. Figure 3 (a) contains an undescribed symbol (white circle in the right corner).
- The references to figures are not provided through the text. e.g. “The distribution of TiAl3 is uniform without agglomeration when the reaction temperature is higher than 1273 K.” and further.
- The phrase “Growing on the three-dimensional dimension” should be modified
- Figure 4. Caption. What was the exposure time?
- What is the sensitivity of the SEM-EDS equipment used in the study? Two decimal places in concentration values (tables in Figures 3,4,5) seem strange.
- “Another point is the enrichment of TiC near TiAl3 and existence of Al4C3, as illustrated by Fig. 6”. Are authors sure that the indicated area in figure 6 corresponds to the Al4C3 phase. Why the particle A has a bright contrast comparing to the matrix, whereas the atomic charge of C is lower than that for Al?
- The grain structure of as-cast Al alloyed with Al-Ti-C and Al-Ti-B are required to confirm the obtained grain size data.
- The text and captions contain multiple technical mistakes.
Reviewer 2 Report
Reviewer’s remarks:
- „These authors claimed: “One of the most effective elements for grain refinement of aluminum alloys is Ti element which forms Ti-containing intermetallic compounds as effective heterogeneous nuclei and has strong growth restriction factor caused by constitutional undercooling [5, 6]”
The authors of the publication rightly attribute the dominant role of Ti as the main component of the Al refining master alloys. However, they limit the State of Art to two groups, that is, Al-Ti-B and Al-Ti-C and the role of the TiB, TiC and TiAl3 phases introduced by these master alloys. The State-of-Art description does not mention role of the ternary aluminides L12 Ti (Al, X) 3, e.g. L12 Ti (Al, Zn) 3 .These phases, due to the similarity of their crystal structure to the α (Al) phase and the small mismatch of the lattice parameter (or rather the nearest-neighbour atomic distance in the relevant contacting planes), constitute an effective substrate for heterogeneous nucleation of the α (Al) phase. It is suggested that in the Introduction they should also provide information about this group of master alloys, the effectiveness of which is described in the relevant literature, e.g.
- P.K. Krajewski, A.L. Greer, W.K. Krajewski, Main Directions of Recent Works on Al-Zn-Based Alloys for Foundry Engineering, Journal of Materials Engineering and Performance 28 (No. 7, 2019) pp. 3086 – 3993
- A.L. Greer, Overview: Application of heterogeneous nucleation in grain-refining of metals, The Journal of Chemical Physics 145 (Issue 21, Dec. 2016), 211704; https://doi.org/10.1063/1.4968846
- In the experimental part, the authors use Ti chips contaminated with vanadium. Contemporary literature provides information on the Al structure refining effect of Al-V phases, e.g.
- Gamal Mohamed Attia Mahran, Abdel-Nasser Mohamed Omran, Grain Refining of Aluminium and 6063 Alloys Using Al-V Alloy Containing Al3V Intermetallic Compound, Materials Science (Medžiagotyra), ISSN 1392–1320, http://dx.doi.org/10.5755/j02.ms.28297
Please note, that in Fig. 2(a) there are unmarked reflections in 2-theta positions around 47.3 or 72.4 degrees. Could these be reflections coming from Al10V (Al21V2 elsewhere) or Al3V phases, which in the {1 1 1} atomic plane may be the substrates of alpha (Al) heterogeneous nucleation. This possibility should be here discussed and the above publication should be also included.
Please note, that the mentioned suggestion of including the above papers to the references list is for genuine scientific reasons of the presenting recent State of the Art in the field: “development of master alloys for Al and Al-base alloys grain refinement”.
- Macrostructures showing grain refining efficiency of the prepared Al-5Ti-0.25C master alloy compared with the Al-5Ti-1B one should be added (see p. 3.4. Refining performance), as well as the method of grain-size measurement should be described in the p. 2. Materials and Methods.
- Examples of the needed text corrections:
- These authors use “ …Ti-containing intermetallic compounds as effective heterogeneous nuclei….. “ or “…TiB2 nuclei..”. In fact, these are nucleant particles or substrates of heterogeneous nucleation, not nuclei.
- page 4, bottom rows: “…by the mothod carefully described in literature [19]” (rather “..by the method described in detail in literature [19]”.
- “strip-like” shape of the TiAl3 particles (rather “needle-like” is used in the relevant literature).
It is suggested the authors use of the professional English technical language check service.
Round 2
Reviewer 1 Report
The manuscript “Preparation process and phase transformation of Al-5Ti-0.25C master alloy adopting Ti machining chip” was significantly improved after revision. I recommend publishing this paper after clarifying following aspects:
- “Extensive studies have shown that the "poisoning" phenomenon of Al-Ti-B master alloys occurs commonly in hypereutectic Al-Si and alloys containing Zr, but Al-Ti-C does not have this defect. We have corrected the problem you raised and described it in detail in the manuscript.” Could the authors clarify what is the "poisoning" and so add this information in introduction.
- “ Al4C3 phase with a small-particle morphology whose altitude is slightly higher than Al substrate plane has a brightness contrast due to the generation of more secondary electrons than the Al substrate.” The images were obtained in secondary electrons? Please, mention this in the caption.
- The scale is too small in Fig.9. I propose the authors to add the inserts in Fig.9 with magnified microstructures. In present state its difficult to estimate the grain size.
Reviewer 2 Report
Dear Aithors
In my opinion the paper can be published in the present, revised form.
Author Response
We are sincerely grateful to the reviewer for your positive affirmation and constructive evaluation for our work.